# Co-Occurrence of Beckwith–Wiedemann Syndrome and Early-Onset Colorectal Cancer

**DOI:** 10.3390/cancers15071944

**Published:** 2023-03-23

**Authors:** Francesco Cecere, Laura Pignata, Bruno Hay Mele, Abu Saadat, Emilia D’Angelo, Orazio Palumbo, Pietro Palumbo, Massimo Carella, Gioacchino Scarano, Giovanni Battista Rossi, Claudia Angelini, Angela Sparago, Flavia Cerrato, Andrea Riccio

**Affiliations:** 1Department of Environmental Biological and Pharmaceutical Sciences and Technologies (DiSTABiF), Università degli Studi della Campania “Luigi Vanvitelli”, 81100 Caserta, Italy; 2Department of Biology, Università degli Studi di Napoli “Federico II”, 80126 Napoli, Italy; 3Division of Medical Genetics, Fondazione IRCCS “Casa Sollievo della Sofferenza”, 71013 San Giovanni Rotondo, Italy; 4Medical Genetics Unit, Azienda Ospedaliera “San Pio” P.”Gaetano Rummo”, 82100 Benevento, Italy; 5Istituto Nazionale Tumori, IRCCS Fondazione G. Pascale, 80131 Napoli, Italy; 6Istituto per le Applicazioni del Calcolo (IAC) “Mauro Picone”, Consiglio Nazionale delle Ricerche (CNR), 80131 Napoli, Italy; 7Institute of Genetics and e Biophysics (IGB) “Adriano Buzzati-Traverso”, Consiglio Nazionale delle Ricerche (CNR), 80131 Napoli, Italy

**Keywords:** Beckwith–Wiedemann syndrome, colorectal cancer, DNA methylation, genomic imprinting, imprinting disorders, CFTR

## Abstract

**Simple Summary:**

Beckwith–Wiedemann Spectrum (BWSp) is a disorder predisposing to tumors of embryonic origin arising during childhood. Little is known concerning tumor risk and histotype prevalence in adult BWSp patients. However, some reports of co-occurrence of BWSp and tumors in early adulthood suggest that the cancer risk in this syndrome may also be relevant later than childhood. Here, we report for the first time a case of co-occurrence of BWSp and early-onset colorectal cancer (EO-CRC). The results demonstrate genetic and epigenetic molecular lesions at both somatic and germline levels, providing support to the hypothesis that epigenetic changes contribute to cancer initiation in tissues where a genetic insult is already present, thus acting in a cooperative manner to stimulate tumorigenesis. This study adds further evidence to the need for tumor surveillance beyond childhood in patients with BWSp.

**Abstract:**

CRC is an adult-onset carcinoma representing the third most common cancer and the second leading cause of cancer-related deaths in the world. EO-CRC (<45 years of age) accounts for 5% of the CRC cases and is associated with cancer-predisposing genetic factors in half of them. Here, we describe the case of a woman affected by BWSp who developed EO-CRC at age 27. To look for a possible molecular link between BWSp and EO-CRC, we analysed her whole-genome genetic and epigenetic profiles in blood, and peri-neoplastic and neoplastic colon tissues. The results revealed a general instability of the tumor genome, including copy number and methylation changes affecting genes of the WNT signaling pathway, CRC biomarkers and imprinted loci. At the germline level, two missense mutations predicted to be likely pathogenic were found in compound heterozygosity affecting the Cystic Fibrosis (CF) gene CFTR that has been recently classified as a tumor suppressor gene, whose dysregulation represents a severe risk factor for developing CRC. We also detected constitutional loss of methylation of the *KCNQ1OT1*:TSS-DMR that leads to bi-allelic expression of the lncRNA *KCNQ1OT1* and BWSp. Our results support the hypothesis that the inherited CFTR mutations, together with constitutional loss of methylation of the *KCNQ1OT1*:TSS-DMR, initiate the tumorigenesis process. Further somatic genetic and epigenetic changes enhancing the activation of the WNT/beta-catenin pathway likely contributed to increase the growth advantage of cancer cells. Although this study does not provide any conclusive cause–effect relationship between BWSp and CRC, it is tempting to speculate that the imprinting defect of BWSp might accelerate tumorigenesis in adult cancer in the presence of predisposing genetic variants.

## 1. Introduction

Beckwith–Wiedemann spectrum (BWSp, OMIM: 130650) is a congenital imprinting disorder (ID) characterized by a wide spectrum of clinical features, including overgrowth, macroglossia, abdominal wall defects, lateralized overgrowth, hypoglycemia and a susceptibility to develop embryonal tumors [1]. At the molecular level, it is caused by defects affecting the imprinted gene cluster located on chromosome 11p15.5. Imprinted genes are characterized by mono-allelic and parent-of-origin-dependent expression that is controlled by regulatory elements displaying differential DNA methylation on the maternal and paternal chromosomes and known as imprinted differentially methylated regions (iDMRs). The 11p15.5 cluster is organized into two regulatory domains: the telomeric domain, including the *H19* and *IGF2* genes, which is controlled by the paternally methylated *H19/IGF2*:IG-DMR (here referred to as IC1); and the centromeric domain, including the *KCNQ1OT1* and *CDKN1C* genes, that is controlled by the maternally methylated *KCNQ1OT1*:TSS-DMR (here referred as IC2) [2]. A molecular diagnosis can be reached for approximately 80% of the BWSp patients. The most common molecular defect of BWSp is a loss of methylation (LoM) at IC2, detected in 50% of cases, followed by mosaic paternal uni-parental disomy (upd) of 11p15 (20%), gain of methylation (GoM) at IC1 (5–10%), intragenic *CDKN1C* mutations (5%) and chromosomal abnormalities in 11p15.5 (less than 5%) [1,3]. Moreover, one-third of the patients with IC2 LoM display multi-locus imprinting disturbances (MLID) affecting more than two imprinted loci on different chromosomes [1].

Up to 8% of all BWSp patients develop an embryonal tumor during childhood [4]. The risk and type of tumor depend on the molecular defect. The highest risk is associated with IC1 GoM, as 28% of cases develop malignancies, mostly nephroblastoma (also known as Wilms tumor). The tumor risk is 16% for the upd (11) subgroup, and nephroblastoma (7.9%), hepatoblastoma (3.5%), neuroblastoma (1.4%) and adrenocortical carcinoma (1.1%) represent the most frequent histotypes. The lowest risk (2.6%) is associated with IC2 LoM, and the related histotypes are hepatoblastoma (0.7%), rhabdomyosarcoma (0.5%) and neuroblastoma (0.5%). Furthermore, in the subset of BWSp patients with MLID, to date, no specific tumor-risk estimates have been defined [5].

Despite the link with embryonal neoplasia, there is no reported association between BWSp and common adult-onset carcinomas, as only rare endocrine tumors have been described in adults with BWSp [1,6,7,8,9,10]. However, a recent study on a small cohort of BWSp patients indicates that the malignancy rate observed in early adulthood reaches that observed in the first decade of life, cumulatively raising the tumor rate to about 20% [11]. Moreover, alterations in imprinted gene expression, methylation or both are frequently found in human cancers, including those of the adult age, such as colorectal cancer (CRC) [12,13,14,15].

CRC has never been reported to be associated with BWSp. A particular case of BWSp who had developed multiple tumors (Wilms tumor, thyroid cancer, and breast cancer) and six colonic polyps has been reported by Fleisher et al., in 2000 [16]. CRC is a common adult-onset carcinoma that accounts for approximately 10% of all annually diagnosed cancers and cancer-related deaths worldwide [17]. Depending on the origin of the mutation, colorectal carcinomas can be classified as sporadic (70%), inherited (5%) and familial (25%). Well characterized molecular mechanisms of CRC are chromosomal instability (CIN), microsatellite instability (MSI) and the CpG island methylator phenotype (CIMP). Within each of these subtypes of CRC, single nucleotide variants (SNVs), copy number variants (CNVs) and chromosomal translocations have been reported to affect critical growth-controlling pathways (e.g., *WNT, MAPK/PI3K, TGF-β, TP53*), and mutations in genes such as *c-MYC, KRAS, BRAF, PIK3CA, PTEN, SMAD2* and *SMAD4* can be used as predictive markers for patient outcomes. In addition to genetic variants, alterations in the expression of ncRNAs, such as lncRNA or miRNA, can also contribute to different steps of the carcinogenesis process and have a predictive value when used as biomarkers [18]. At diagnosis, the median age of patients with CRC is 68 and 72 years in men and women, respectively [19]. However, around 5% of all CRCs are diagnosed in patients < 45 years old [20]. Among EO-CRC cases, approximately 30% of patients are affected by tumors harboring mutations causing hereditary-cancer-predisposing syndromes, and 20% have familial CRC. The remaining 50% of EO-CRC patients have neither hereditary syndromes nor familial CRC and are classified as sporadic [20].

Here, we report on a young woman affected by BWSp who developed CRC at age 27. Genome-wide epigenetic and genetic analyses were performed to look for a possible link between these two rare pathologic events.

## 2. Material and Methods

### 2.1. Ethics

Genetic analyses were performed after written informed consent was obtained from the patient. The research work was carried out according to ethical principles and the Italian legislation. The study was approved by the Ethical Committee of the University of Campania “Luigi Vanvitelli” (Naples, Italy. Approval Number:1135, 13 October 2016).

### 2.2. Tumor Diagnostics

The expression of the mismatch repair (MMR) proteins (MLH1, MSH2, MSH6 and PMS2) associated with microsatellite instability of CRC was evaluated by immunohistochemistry using monoclonal antibodies on formalin-fixed and paraffin-embedded (FFPE) tumor samples. Mutation analysis of *KRAS*, *NRAS* and *BRAF* genes was performed by using the “Oncomine solid tumor” panel on the NGS-Ion Torrent platform (after DNA extraction from FFPE tumor samples and according to the protocols of manufacture (ThermoFisher Scientific, Monza, Italy).

### 2.3. Genetic and Epigenetic Analysis of Blood, Neoplastic and Perineoplastic Tissues

*DNA extraction*. Genomic DNA was extracted from peripheral blood leukocytes (PBL), neoplastic tissue and peri-neoplastic tissue by the salting-out procedure, and its concentration was determined using a NanoDrop spectrophotometer (ThermoFisher Scientific, Italy).

### 2.4. DNA Methylation Analysis

Methylation-Specific Multiple-Ligation-Dependent Probe Amplification (MS-MLPA) was performed on 50 ng of PBL genomic DNA of the patient and three non-affected individuals, in order to analyze DNA methylation and CNVs of several iDMRs using the SALSA MS-MLPA Kit ME034-B1 (MRC-Holland, Amsterdam, The Netherlands). The amplified products were separated by capillary electrophoresis using ABI 3500 Genetic Analyzer (Applied Biosystems, CA, USA). Data were processed by Coffalyser software (MRC-Holland, Amsterdam, The Netherlands).

Pyrosequencing analysis was carried out as previously reported [21]. The primers for the pyrosequencing assays are reported in Appendix A.

Genome-wide methylation analysis was performed by the Illumina Infinium methylation Epic array on bisulfite-converted DNA extracted from the three tissues of the patient. Data were analyzed using R version 4.1.0. Beta values (Bvalues) were derived from “idat” files by the “champ.load” module of the “ChAMP” R package (v.2.22.0), with quality control options set as default and array type as “EPIC.” To adjust the Bvalues of type 2 probes, we applied BMIQ normalization with the default options for array type as “EPIC.” The iDMR coordinates were found on the web site http://www.humanimprints.net/ (accessed on 7 November 2022). The methylation profile of each iDMR was calculated as the average of the CpG methylation levels within the iDMR. PBL methylation values of the patient were compared with those of 8 age-matched controls of the same batch downloaded from GSE195873 (C5 to C12) [22]. The values, deviating ± 3 s.d. and differing at least 10% from the mean of the controls, were classified as abnormal. The pathways analysis of the tumor tissue was performed using gProfiler2 package (v0.2.0) on the DMGs (differentially methylated genes), defined as genes with at least one CpG with +/−30% methylation compared to the peri-neoplastic tissue. We used the CpG island coordinates downloaded by the UCSC genome browser to calculate the mean methylation value of both the enriched genes pathway and the methylation-based biomarkers with a Dbeta of +/−20%. The raw and processed files are deposited in GEO under the accession GSE224222.

### 2.5. Copy Number Analysis

Whole-genome copy number variants (CNVs) were analysed by single nucleotide polymorphism (SNP)-array that was carried out as previously reported [22,23].

### 2.6. DNA Sequencing

Whole-genome single nucleotide variants were analysed by *Whole Exome Sequencing (WES)* as previously described [22,24]. In silico prediction of variant pathogenicity on the CFTR protein was studied by Polyphen-2 (http://genetics.bwh.harvard.edu/pph2/ (accessed on 11 January 2023) and following the guidelines of the American College Medical Genetics (ACMG) reported in [25].

Primers for PCR amplification and Sanger sequencing of CFTR variants segregation analysis are reported in Appendix A.

### 2.7. Expression Analysis

RNA was extracted from neoplastic and peri-neoplastic tissues using TRIZOL reagent (Invitrogen, ThermoFisher Scientific, Italy) according to manufacturer’s protocol. The concentration was determined using a NanoDrop spectrophotometer (ThermoFisher Scientific, Italy). About 1.5 µg of RNA was treated with RNase-free DNase (Sigma-Aldrich, St. Louis, Missouri, Stati Uniti), and first-strand complementary DNA (cDNA) was synthesized using QuantiTect Reverse Transcription Kit (Qiagen-Italia, Milan, Italy) following the manufacturer’s protocol.

Quantitative PCR (qPCR) analysis was performed in a StepOnePlus RT-qPCR system (Applied Biosystem). Each reaction was carried out in duplicate in a 20 µL final volume with 10 µL of SYBR-Green supermix, 0.5 µM of each primer and 5µL of cDNA. The initial denaturation for each run was for 10 min at 95 °C. This was followed by 40 amplification cycles of 95 °C for 15 s, and for 1 min at 60° followed by 30 s at 72 °C. Relative expression levels were normalized using *GAPDH* as a reference gene. Primers are listed in Appendix A. Allele-specific expression analysis of *KCNQ1OT1* was performed by amplifying about 100 ng of cDNA. Sequencing of the SNP rs1076621 allowed us to distinguish the two alleles. PCR primers used were the same as those for Real Time-qPC (RT-qPCR).

## 3. Results

The young woman described in this study was referred to the epigenetic laboratory of Vanvitelli University at the age of 15 years with suspected clinical diagnosis of BWSp for the presence of ear pits and heterometry of the upper and lower limbs (clinical score for BWSp = 3). The clinical diagnosis of BWSp was confirmed by the finding of IC2 LoM in PBL DNA by bisulphite treatment, followed by amplification and pyrosequencing (Appendix A). At the age of 27 years, the patient developed a rectal carcinoma that was treated surgically and analysed at the pathology department of Fondazione Pascale hospital, in Naples. The reported histological diagnosis was: rectal mucosa showing neoplastic proliferation with features of invasive adenocarcinoma composed of 60% tumor cells. Immunohistochemical analysis demonstrated normal expression of the mismatch repair (MMR) proteins MLH1, MSH2, MSH6, and PMS2 in the nuclei of the neoplastic cells. Moreover, mutational analysis of the *KRAS*, *NRAS* and *BRAF* genes did not reveal any pathogenic variant in the neoplastic tissue. The proband’s parents and siblings did not show any sign of BWSp nor did they report any history of CRC.

Due to the extremely rare co-occurrence of BWSp and CRC in a young individual, further molecular analyses were performed to identify a possible link between the etiologies of these two diseases. First, a molecular test for MLID was performed on PBL DNA by MS-MLPA to detect methylation and copy number changes at 10 disease-associated imprinted loci. No further methylation alteration other than IC2 LoM and no CNV were detected at any of the tested imprinted loci with this assay (Figure 1a). To extend the methylation analysis to additional iDMRs, a whole-genome approach was employed by using the Illumina Infinium methylation EPIC array, and the results were compared with the methylation profiles of 8 age-matched controls (Figure 1b). Among the newly analysed DMRs, the *ERLIN2*:Int6-DMR and *MKRN3*:TSS-DMR were found to be hypomethylated as their methylation level was 22% and 24% lower than the mean of controls, respectively (Figure 1b and Appendix A).

According to the recently proposed definition considering MLID as only those cases with imprinting defects at (a) ≥ 1 clinically relevant loci and (b) ≥ 2 additional (germline) ICs [26], we conclude that our patient has mild MLID in her PBL DNA.

Because of the possible link between IC2 LoM and EO-CRC, we analysed the expression of *KCNQ1OT1, KCNQ1* and *CDKN1C* in the neoplastic and peri-neoplastic tissues (Figure 2). We found that the total RNA level of all the three genes as determined by RT-qPCR was increased in the neoplastic tissue if compared to the peri-neoplastic tissue, although their expression was relatively low (Figure 2a). We then tested the allelic expression of *KCNQ1OT1* by using a SNP present in the transcribed region and found that this gene was transcribed on both parental alleles with a similar ratio in the neoplastic and peri-neoplastic tissues (Figure 2b). Thus, IC2 methylation and *KCNQ1OT1* imprinting were similarly relaxed in the neoplastic and peri-neoplastic tissues and *KCNQ1* was increased in the tumor of our patient.

Genome-wide methylation analysis was then performed on the neoplastic and peri-neoplastic colon tissues. Only the CpGs showing methylation values in cancer that were at least 30% lower or higher than those of the peri-neoplastic tissue were considered. Furthermore, by filtering for the probes of the CpG islands overlapping promoter regions, 1042 genes were identified as differentially methylated genes (DMGs). These genes were annotated, and the Kyoto Encyclopedia of Genes and Genomes (KEGG) pathway enrichment was calculated by g:profiler tool (biit.cs.ut.ee/gprofiler). The KEGG pathway analysis revealed four main biologically significant and enriched processes in the cancer tissue: the Neuro ligand–receptor interaction, the Cell adhesion molecules, the Calcium signaling and the WNT signaling pathways (Table 1).

Term size: number of the genes involved in each pathway; intersection size: number of the hypo/hyper DMGs significantly enriched in each pathway.

We then focused on the DMGs whose CpG island overlapping the promoter region had a +/−20% fold change of mean methylation values in the tumor with respect to the peri-neoplastic tissue. Interestingly, all these DMGs were hypermethylated including the methylation-based CRC biomarkers *SFPR1*, *SFPR2* and *WIF1* (Figure 3a and Appendix A). We then looked at the methylation of further CRC biomarker candidates [27] and found that the promoters of the *BMP3*, *CMTM3*, *CNRIP1* and *MDF* genes were hypermethylated as well (Figure 3b and Appendix A). In contrast, the MMR genes revealed no methylation change at their promoters, consistent with the results of the immunohistochemical analysis (Appendix A). In addition, 8 CIMP-specific genes revealed no methylation change at the CpG islands-promoters; thus, this tumor was considered CIMP-negative (Appendix A) [28].

DNA methylation of the iDMRs in the neoplastic and peri-neoplastic tissue was also analysed (Figure 2c and Appendix A). The mean methylation values of the *PPIEL*, *MEST, PEG13, RB1, NDN, GNAS-AS1, GNAS-XL* and *GNAS-A/B* DMRs were found to be at least 10% higher in the neoplastic than in the peri-neoplastic tissue (Figure 3c and Appendix A). Conversely, the *DIRAS3, GPR1-AS, ZNF597, ZNF331, GNAS-NESP, WRB* and *SNU13* DMRs were shown to be at least 10% hypomethylated. These results were confirmed by sodium bisulfite treatment, amplification and pyrosequencing (Appendix A).

A SNP-array analysis was performed to investigate if any of these methylation changes were associated with copy number variants. No relevant CNV and loss of heterozygosity (LOH) were detected in PBL and peri-neoplastic tissues (Appendix A). Conversely, several mosaic and non-mosaic CNVs affecting tumor-associated genes and several imprinted loci were found in the neoplastic tissue (Figure 4 and Appendix A). In particular, non-mosaic duplications and microduplications affected the long arm of chrs 8 and 20 and segments of several chromosomes involving the oncogenes *cMYC* (chr8q24.21), *KRAS* (chr 12p12.1) and *AURKA* (20q13.2) and the imprinted loci *GNAS* (chr 20q), *PEG13* and *ERLIN2* (chr 8q). Despite the CNV, no further methylation defect of *ERLIN 2* was detectable in the tumor (Figure 3c), likely because of the constitutional hypomethylation affecting this DNA region. Numerous mosaic CNVs were also revealed, including deletions of the whole chromosome 9, long arm of chrs 8 and 18, short arm of chr 17 and segments of chrs 4q and 12p, overall affecting several tumor suppressor genes, such as *TPP53* (17p13.1), *SMAD2* (18q21.1), *SMAD4* (18q21.2), *DCC* (18q21) and the WNT antagonist *SFRP2* (4q31.3) (Figure 4 and Appendix A). No CNVs or LOH involving the MMR complex genes were found.

Overall, the methylome and SNP array results demonstrate a general instability of the tumor genome, including copy number and methylation changes (both primary and CNV-dependent) that affected genes of the Neuro ligand–receptor interaction, Cell adhesion molecules, WNT and Calcium signaling pathways, CRC biomarkers and imprinted loci.

To investigate the presence of germline genetic variants predisposing to CRC, we performed whole-exome sequencing of PBL DNA of the patient. Two very rare SNVs of the cystic fibrosis transmembrane conductance regulator (*CFTR*) gene were found in heterozygosity in the patient genome. Both variants were transversions at the nucleotide level and missense mutations at the amino acid level, and were classified as possibly damaging by PolyPhen-2 and likely pathogenic according to the ACMG criteria (Figure 5a). The variant c.3154T > G, p.Phe1052Val (rs150212784) affected an amino acidic residue of the ABC transmembrane type-1 2 protein domain (Figure 5a,b). The variant c.4054C > G, p.Gln1352Glu (rs751098333) affected an amino acidic residue of the ABC transporter 2 domain. Sanger sequencing analysis of the trio demonstrated that the two CFTR variants were present in compound heterozygosity in the patient, as the p.Gln1352Glu was inherited from the mother and the p.Phe1052Val was inherited from the father (Figure 5c).

## 4. Discussion

BWSp is a disorder predisposing to tumors of embryonic origin that arise during childhood, usually up to 10 years of age [4,5,27]. Data on tumor risk and histotype prevalence in adult BWSp patients are very limited because the clinical phenotype mitigates with age and a follow-up of large series of adults with BWSp has not been performed [1,11]. Therefore, whether BWSp increases cancer risk in adulthood remains an open question. To look for a possible relationship between the etiologies of BWSp and EO-CRC in this patient, we investigated her molecular lesions at both somatic and germline levels.

Neither the loss of MMR proteins nor pathogenic variants of *KRAS*, *NRAS* and *BRAF*, which are molecular lesions usually associated with hereditary or familial EO-CRC [20], were revealed in the neoplastic colon tissue. These results are consistent with a sporadic form of EO-CRC and suggest that the microsatellite instability pathway is not involved in the onset and development of cancer in our patient. Normal methylation of the promoters of 8 CIMP-specific genes reveals the CIMP negative status of the tumor. In contrast, the SNP-array data revealed a general chromosomal instability, as numerous mosaic and non-mosaic CNVs were detected, including mosaic deletions of arms 8p, 17p and 18q, and amplification of the 20q11.21, which are frequently found in CRCs characterized by CIN (Figure 3 and Appendix A) [29,30]. The methylome analysis revealed a general epigenome instability of the neoplastic tissue, showing primary or CNV-dependent methylation changes across the whole genome and involving tumor-associated genes and imprinted loci (Figure 3 and Appendix A). Notably, several DMGs belonged to the WNT signaling, Cell adhesion, Neuroactive ligand-receptor interaction and Calcium signaling pathways, in accordance with previously reported CRC methylome data [31,32]. These pathways have all been reported to play a crucial role in the onset of CRC or in driving its progression [18,32,33]. Particularly important is the WNT pathway, whose dysregulation leads to aberrant cell growth and stem cell differentiation. It also impairs cellular adhesion, favoring migration and metastasis [34]. In addition, human and mouse studies demonstrated that the CNVs and methylation changes found at imprinted loci, such as amplification of *GNAS*, hypermethylation of *MEST/PEG1*, *PEG13* and *NDN*, involved genes activating the WNT pathway [35].

At the germline level, two missense mutations, predicted to be likely pathogenic, of the Cystic Fibrosis (CF) gene *CFTR,* were found in compound heterozygosity (Figure 5). Recent studies classify *CFTR* as a tissue-specific tumor suppressor gene, whose inactivation represents a severe risk factor for developing early aggressive CRC in either affected CF patients and healthy carriers [36]. Further, lowered expression of CFTR is reported in sporadic CRC, where downregulation of CFTR is associated with poor survival. The mechanisms underlying the action of CFTR as a tumor suppressor are not clearly understood. However, dysregulation of WNT/β-catenin signaling and disruption of intestinal stem cell homeostasis and intestinal barrier integrity, as well as intestinal dysbiosis, immune cell infiltration, stress responses and intestinal inflammation, have all been reported in human CF patients and in animal models [37].

Patients with CF show a high incidence of early aggressive CRC beginning in their 30s, with 50% of CF patients having developed tumors by the age of 40 [38]. Our patient developed the carcinoma even earlier, suggesting that further molecular alterations might trigger the tumorigenesis in addition to CFTR mutations. The only other constitutional molecular lesion found in this patient is IC2 LoM leading to bi-allelic expression of the lncRNA *KCNQ1OT1* and BWSp. Intriguingly, several studies indicate that loss of imprinting and *KCNQ1OT1* overexpression plays an important role in CRC carcinogenesis. In particular: (i) increased expression of *KCNQ1OT1* has been reported in CRC tissue and cell lines [39]; (ii) *KCNQ1OT1* is a prognostic biomarker correlating with poor prognosis of CRC patients [40]; and (iii) *KCNQ1OT1* promotes proliferation, migration and invasion of CRC cells [41]. It has been demonstrated that β-catenin can induce *KCNQ1OT1* transcription through direct binding to its promoter and may contribute to CRC development by functioning as a lncRNA regulatory factor [42]. Additionally, the lncRNA *KCNQ1OT1* regulates the imprinted expression of the *KCNQ1* and *CDKN1C* genes, both showing tumor suppressor activity in several tumors, including CRC. *KCNQ1* encodes a potassium channel protein subunit and represents an early prognostic biomarker in CRC, whose deficiency is associated with poor outcomes [43]. It has been demonstrated that KCNQ1 limits oncogenesis by inhibiting nuclear localization of β-catenin and by maintaining adherent junctions that prevent the epithelial to mesenchymal transition [44]. *CDKN1C* is an important inhibitor of the cell cycle, whose loss of expression appears associated with colorectal carcinogenesis and poor prognosis [45,46]. However, it is important to note that IC2 methylation and *KCNQ1OT1* imprinting were similarly relaxed in the neoplastic and peri-neoplastic tissues, and *KCNQ1* and *CDKN1C* were not repressed in the tumor with respect to normal tissue in our patient. (Figure 2b and Appendix A). Methylation mosaicism among tissues and the presence of non-tumor cells in the tumor biopsy may in part explain this result.

Thus, our findings suggest that the rare combination of a genetic and an epigenetic lesion might have contributed to cause a very EO-CRC by triggering beta-catenin activation. We hypothesize that in our case, the germline CFTR mutations represented the first hit of the tumorigenesis process and that this is accelerated by IC2 LoM and/or MLID. Further somatic events, including epigenetic silencing of WNT antagonist genes, might have further enhanced activation of the WNT/beta-catenin pathway and provided a growth advantage to cancer cells.

Our study adds further evidence to the hypothesis that epigenetic changes are necessary for cancer initiation in tissues where a genetic insult is already present, acting in a cooperative manner to stimulate tumorigenesis [47]. However, further studies are needed to establish whether the BWSp epimutations represent an increased risk factor for adult cancer.

## 5. Conclusions

In conclusion, we described the first case of co-occurrence of BWSp and EO-CRC. We found IC2 LoM as cause of BWSp, likely pathogenic *CFTR* variants as possible germline lesions predisposing to CRC, and a chromosomal and epigenome instability as an altered molecular profile of the tumor genome. The development of a very EO-CRC can be explained by the adverse circumstance of the co-occurrence of CFTR mutations and IC2 LoM, likely responsible for cancer initiation and further triggered by somatic events enhancing the WNT/beta-catenin pathway. From a clinical point of view, our study supports the need to extend cancer surveillance beyond the interval normally recommended for BWSp patients.

## Figures and Tables

**Figure 1 cancers-15-01944-f001:**
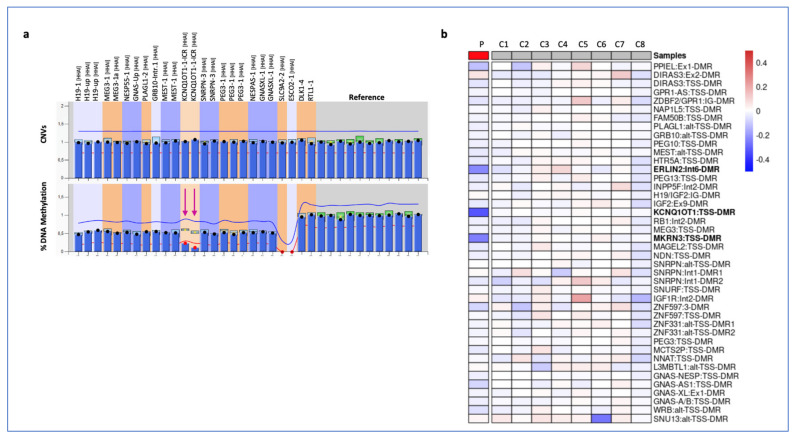
Methylation analysis of the iDMRs on PBL DNA of the proband and control subjects by (**a**) multiple MS-MLPA and (**b**) methylation-array. (**a**) Copy Number (CNVs, upper panels) and DNA methylation (lower panels) were analysed at 10 imprinted loci and compared with the mean values of three controls to determine the relative copy number and methylation percentage. The arrows indicate the iDMRs affected by abnormal methylation. (**b**) Heat map showing methylation values of 42 iDMRs of the proband (P) normalized against the mean of eight controls (C1-C8). The iDMRs in bold showed values with a fold change ≤ 3 standard deviation and ≤ 10% of the mean of controls.

**Figure 2 cancers-15-01944-f002:**
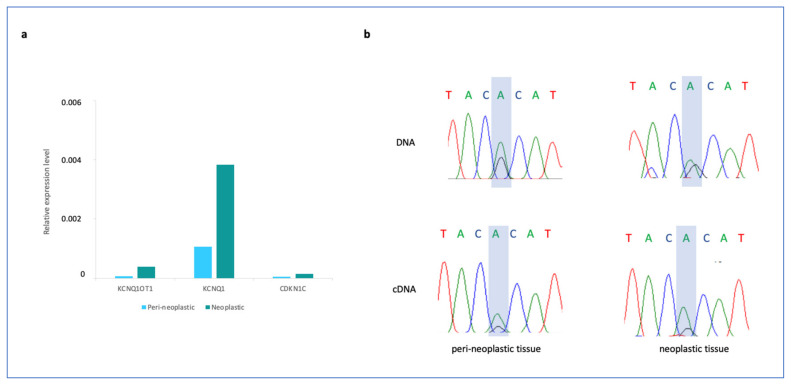
Expression analysis of *KCNQ1OT1*, *KCNQ1* and *CDKN1C* in the neoplastic and peri-neoplastic tissues. (**a**) Total expression levels of the three genes obtained by RT-qPCR and (**b**) allele-specific expression levels of *KCNQ1OT1* obtained by Sanger sequencing following Reverse transcriptase-PCR. In (b) the rectangle indicates the polymorphism to distinguish the two alleles.

**Figure 3 cancers-15-01944-f003:**
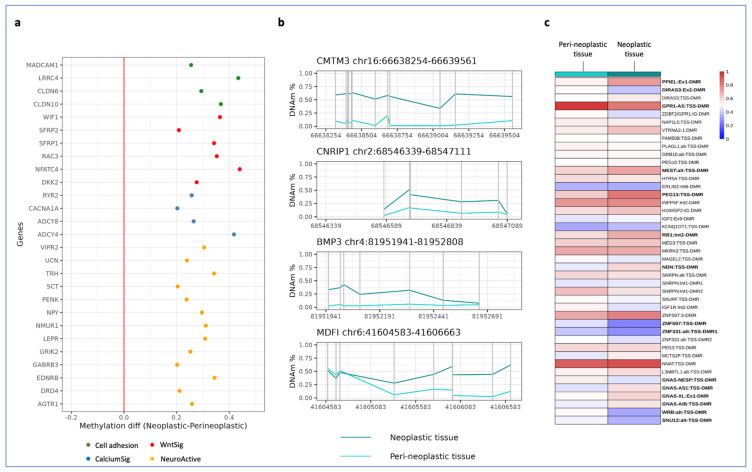
Genome-wide methylation analysis of neoplastic tissue compared with peri-neoplastic tissue. (**a**) DMGs involved in the four identified KEGG pathways. All of the DMGs showed hypermethylation at the promoter region. (**b**) Four biomarkers identified among the DMGs. Each vertical line represents the genome coordinate of a single probe in the array (**c**) Heat map showing the methylation profile of the iDMRs in neoplastic and peri-neoplastic tissues. The iDMRs with a mean methylation difference of at least 10% between neoplastic and peri-neoplastic tissues are indicated in bold. Methylation changes possibly present in the peri-neoplastic tissue could not be detected as no normal colorectal tissue specimen was available for normalization.

**Figure 4 cancers-15-01944-f004:**
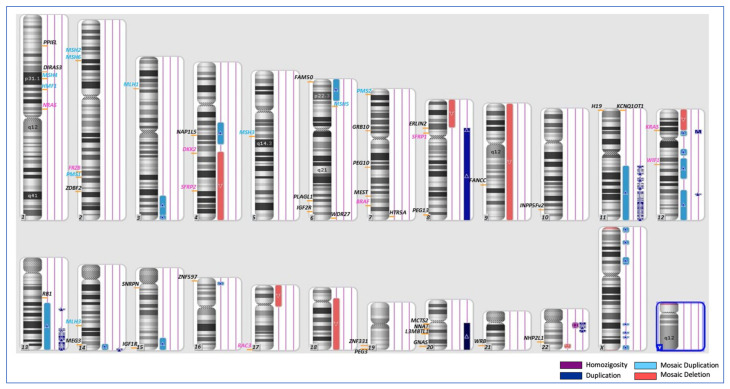
Karyoview of tumor tissue analyzed by SNP-Array. The vertical bars represent: regions of homozygosity (purple), duplications (dark blue), mosaic duplications (light blue) and mosaic deletions (red). Names of genomic loci are shown in black for iDMRs, in pink for genes involved in carcinogenesis and in light blue for MMR complex genes.

**Figure 5 cancers-15-01944-f005:**
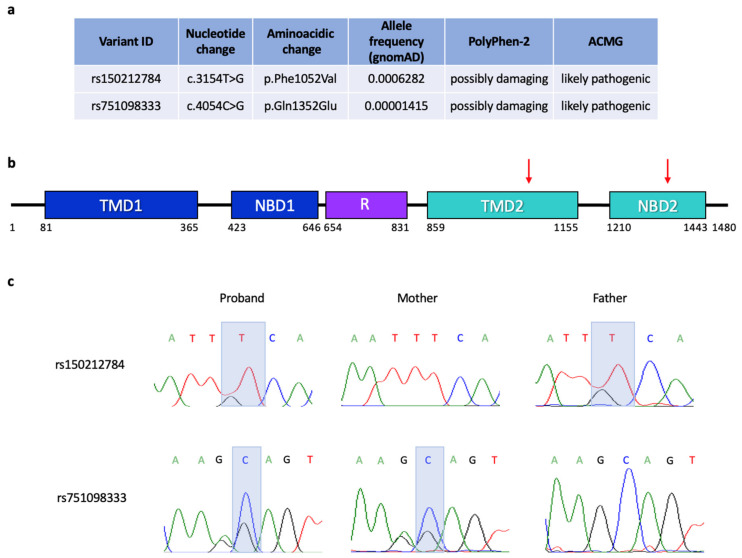
Characteristics of the *CFTR* variants identified in the patient genome. (**a**) Scheme showing the ID, nucleotide and amino acid variations, frequency and prediction on the protein function of the two variants. (**b**) Domain structure of human CFTR protein. TMD: transmembrane domain, NBD: nucleotide-binding domain, R: regulatory domain. The position of the two missense mutations is indicated by red arrows. (**c**) Electropherograms of the Sanger sequencing results obtained on the trio, validating the variants detected by WES and showing their segregation in the family. The grey rectangles highlight the picks of the nucleotide variants.

**Table 1 cancers-15-01944-t001:** KEGG pathway analysis of the DMGs.

Term_Name	Term_Size	Intersection_Size	*p*_Value
Neuroactive ligand–receptor interaction	227	44	2.137936^−9^
Calcium signaling pathway	198	30	8.469467^−4^
WNT signaling pathway	155	23	1.356191^−2^
Cell adhesion molecules	104	17	3.260210^−2^

## Data Availability

Methylation array data generated and analyzed during the current study have been deposited under accession code GSE224222 in the Gene Expression Omnibus repository.

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
