# Peer review of "Co-Occurrence of Beckwith–Wiedemann Syndrome and Early-Onset Colorectal Cancer"

_cancers, 2023, doi:10.3390/cancers15071944_

Round 1
Reviewer 1 Report
The study is well-designed and the experiments were performed correctly. They reported interesting findings including mutations in the CF gene and constitutional loss of methylation 45 of the KCNQ1OT1:TSS-DMR that leads to biallelic expression of the lncRNA KCNQ1OT1 and 46 BWSp. Although based on the reported data it is virtually hard to make a true link between BWS and CRC, the data could provide a foundation for further research in similar cases.
Minor comments:
1. Based on the findings of this study how would the author comment on whether the patients with BWS should have more intensive colorectal cancer screening than the general population?
2. The mutations identified in the CF gene are likely pathogenic by ACMG as stated in Fig. 4, but in lines 43, 317, and 360 they are regarded as pathogenic germ-line mutations. How this conclusion was made?
3. Missing the article by Fleisher et al, in GASTROENTEROLOGY 2000;118:637.
4. More explanations are needed to Fig. 1 data.
5. In Fig. 4, the abbreviations should be explained in the legend.
6. In Fig2a four pathways were depicted but the legend refers to two pathways, missing one. needs explanation?
Author Response
Point-by-point response to the reviewers’ comments and a description of the changes made in the manuscript CANCERS-2218673.
We thank the reviewers for their valuable comments on the original manuscript, CANCERS-2218673, “Co-occurrence of Beckwith–Wiedemann Syndrome and early-onset colorectal cancer”.
We have carefully reviewed the suggestions/concerns of the reviewers and addressed such points. Below we report point-by-point responses (in red) to the reviewers’ comments, and a description of the changes made in the manuscript (underlined).
REVIEWER# 1
The study is well-designed and the experiments were performed correctly. They reported interesting findings including mutations in the CF gene and constitutional loss of methylation of the KCNQ1OT1:TSS-DMR that leads to biallelic expression of the lncRNA KCNQ1OT1 and BWSp. Although based on the reported data it is virtually hard to make a true link between BWS and CRC, the data could provide a foundation for further research in similar cases.
We thank the reviewer #1 for recognizing the value of our manuscript and for the useful comments.
Minor comments:
1.Based on the findings of this study how would the author comment on whether the patients with BWS should have more intensive colorectal cancer screening than the general population?
R#1-1 According to the last international BWSp consensus (Brioude et al., 2018) there is no reported association between BWSp and common adult-onset carcinomas. However, data on cancers in adult BWS patients is limited as reported by Gazzin et al (2019): As tumor risk and overall clinical surveillance are limited to childhood, scientific reports and medical knowledge concerning BWSp is mostly limited to the first decade of life. Very few are currently known on BWSp natural history and presentation in adulthood: information on BWSp impact later in life is limited and rarely reported.
Since statistical data on adulthood cancer risk in BWSp are lacking and increasing number of studies describe case reports affected by BWSp and developing embryonic or adult tumors in adolescence and adult age (ref. 6-11), we believe that a general cancer screening should be included in the clinical management of BWSp patients also in adulthood. Further studies on adulthood cohorts of BWSp will help evaluate tumor histotypes for thorough screening. Please, see the last sentence of the Conclusions.
2.The mutations identified in the CF gene are likely pathogenic by ACMG as stated in Fig. 4, but in lines 43, 317, and 360 they are regarded as pathogenic germ-line mutations. How this conclusion was made?
R#1-2 It is a mistake. We modified the text by replacing “pathogenic” with “likely pathogenic”.
- Missing the article by Fleisher et al, in GASTROENTEROLOGY 2000;118:637.
R#1-3 We have mentioned this article in the text at line 101 of the Introduction by adding the following sentence: “CRC has never been reported associated with BWS, so far. A particular case of BWS who had developed multiple tumors (Wilms tumor, thyroid cancer, and breast cancer) and six colonic polyps has been reported by Fleisher et al in 2000. CRC is a common adult-onset carcinoma that accounts for approximately 10%...”.
- More explanations are needed to Fig. 1 data.
R#1-4 We have better explained the data of MS-MLPA and methylation-array performed on PBL DNA and showed in Fig.1 by modifying the Results paragraph at line 229 as follows:
“First, a molecular test for MLID was performed on PBL DNA by MS-MLPA to detect methylation and copy number changes at 10 disease-associated imprinted loci. No further methylation alteration other than IC2 LoM and no CNV were detected at any of the tested loci (Figure 1a). To extend the methylation analysis to additional iDMRs, a whole-genome approach was employed by using the Illumina Infinium methylation EPIC array, and the results were compared with the methylation profiles of 8 age-matched controls (Figure 1b). Among the newly analysed DMRs, the ERLIN2:Int6-DMR and MKRN3:TSS-DMR resulted to be hypomethylated as their methylation level was 22% and 24% lower than the mean of controls, respectively (Figure 1b and Table S3). According to the recently proposed definition considering MLID only those cases with imprinting defects at (A)≥1 clinically relevant loci and (B)≥2 additional (germline) ICs [26], we conclude that our patient has mild MLID in her PBL DNA.
- In Fig. 4, the abbreviations should be explained in the legend.
R#1-5 The figure 4 (now 5) has been slightly modified and the protein domains are now depicted as reported in UniProt web-site (https://www.uniprot.org/uniprotkb/P13569/entry#family_and_domains) to avoid copy-right issues. The abbreviations have been explained in the legend as follows:
“(b) Domain structure of human CFTR protein. TMD: transmembrane domain, NBD: nucleotide-binding domain, R: regulatory domain. The position of the two missense mutations is indicated by red arrows.”
- In Fig2a four pathways were depicted but the legend refers to two pathways, missing one. needs explanation?
R#1-6 The legend has been corrected as follows: “DMGs involved in the four identified KEGG pathways”
Reviewer 2 Report
In this manuscript, Cecere et al. reported a case of co-occurrence of BWSp and early-onset colorectal cancer (EO-CRC). The authors performed several epigenetic and genetic analyses on PBL, peri-neoplastic tissue, and neoplastic tissue of the patient. In the PBL, the authors found IC2 LoM and compound heterozygous for missense variants of CFTR, which was a causative gene for cystic fibrosis and a tissue-specific tumor suppressor gene. In the neoplastic tissue, the authors found genetic and epigenetic instabilities, which were frequently found in CRC. In addition, the authors found abnormal methylation at several iDMRs. This is a very interesting case and has a possibility, which the rare combination of a genetic and an epigenetic lesion may contribute to cause an EO-CRC. However, as the authors indicated, this study does not provide any conclusive cause-effect relationship between BWS and CRC, and a pure coincidence is not ruled out.
Major comments
1. It is important to show the relationship between the CFTR variants and the IC2 LoM. The allelic and quantitative expression of KCNQ1OT1 and KCNQ1 in the peri-neoplastic and neoplastic tissues should be analyzed. Did the immunohistochemical analysis show nuclear localization of beta-catenin in the neoplastic tissues? In addition, the allelic and quantitative expression of CDKN1C in these tissues should also be analyzed.
2. Since the IC2 LoM is usually mosaic, mosaic ratios in the PBL and the peri-neoplastic tissue may be different. In fact, methylation levels were 14% in the PBL and 0.33 (33%) in the peri-neoplastic tissue. The authors hypothesized that the germline CFTR mutations was the first hit and the IC2 LoM was the second hit of the tumorigenesis. If his is the case, almost all tumor cells show the IC2 LoM, and methylation level of the neoplastic tissue may depend on percentage of tumor cells. However, the methylation levels at IC2 in the peri-neoplastic and neoplastic tissues were not so different. The authors should show the percentage of tumor cells in the neoplastic tissue, which they analyzed, and explain these methylation statuses.
3. In the peri-neoplastic tissue, many iDMRs showed abnormal methylation. How the authors defined and found these abnormal methylations and why these abnormal methylations occurred in the peri-neoplastic tissue?
Minor comments
1. Detailed pathological information of the peri-neoplastic and neoplastic tissues should be described.
2. As the authors mentioned, CIMP was a well characterized molecular mechanism of CRC. It would be better to present the CIMP data.
3. In the legend of Figure 4b, the domain names of CFTR protein should be spelled out.
4. In Discussion, the author should discuss CDKN1C as well as KCNQ1OT1/KCNQ1.
5. A SNP-array analysis revealed that non-mosaic duplications affected the imprinted loci of GNAS (chr 20q), PEG13 and ERLIN2 (chr 8q) in the neoplastic tissue. Although methylation levels of GNAS and PEG13 were altered, that of ERLIN2 was not altered (Figure 2C and Table S3) . The authors should explain these findings.
6. It would be better to add pyrosequencing data of all iDMRs analyzed in the colorectal tissues to Figure S2.
Author Response
Point-by-point response to the reviewers’ comments and a description of the changes made in the manuscript CANCERS-2218673.
We thank the reviewers for their valuable comments on the original manuscript, CANCERS-2218673, “Co-occurrence of Beckwith–Wiedemann Syndrome and early-onset colorectal cancer”.
We have carefully reviewed the suggestions/concerns of the reviewers and addressed such points. Below we report point-by-point responses (in red) to the reviewers’ comments, and a description of the changes made in the manuscript (underlined).
REVIEWER# 2
In this manuscript, Cecere et al. reported a case of co-occurrence of BWSp and early-onset colorectal cancer (EO-CRC). The authors performed several epigenetic and genetic analyses on PBL, peri-neoplastic tissue, and neoplastic tissue of the patient. In the PBL, the authors found IC2 LoM and compound heterozygous for missense variants of CFTR, which was a causative gene for cystic fibrosis and a tissue-specific tumor suppressor gene. In the neoplastic tissue, the authors found genetic and epigenetic instabilities, which were frequently found in CRC. In addition, the authors found abnormal methylation at several iDMRs. This is a very interesting case and has a possibility, which the rare combination of a genetic and an epigenetic lesion may contribute to cause an EO-CRC. However, as the authors indicated, this study does not provide any conclusive cause-effect relationship between BWS and CRC, and a pure coincidence is not ruled out.
Major comments
- It is important to show the relationship between the CFTR variants and the IC2 LoM. The allelic and quantitative expression of KCNQ1OT1 and KCNQ1 in the peri-neoplastic and neoplastic tissues should be analyzed. Did the immunohistochemical analysis show nuclear localization of beta-catenin in the neoplastic tissues? In addition, the allelic and quantitative expression of CDKN1C in these tissues should also be analyzed.
R#2-1 We have analysed the allelic and quantitative expression of KCNQ1OT1 and quantitative expression of KCNQ1and CDKN1C. We found that the total RNA level of all the three genes was increased in the neoplastic tissue if compared to the peri-neoplastic tissue, although KCNQ1OT1 and CDKN1C expression was very low. Concerning the allelic analysis, the results showed that KCNQ1OT1 was expressed on both parental alleles with similar allelic ratio in the neoplastic and peri-neoplastic tissues. Unfortunately, the pathology department did not perform the immunohistochemical analysis of beta-catenin on the bioptic sample. Thus, this information could not be obtained.
To comment these results, we added:
Results line 291: “Because of the possible link between IC2 LoM and CRC, we analysed the expression of KCNQ1OT1,KCNQ1 and CDKN1C in the neoplastic and peri-neoplastic tissues (Figure 2). We found that the total RNA level of all the three genes as determined by RT-qPCR was increased in the neoplastic tissue if compared to the peri-neoplastic tissue, although their expression was relatively low (Figure 2a). We then tested the allelic expression of KCNQ1OT1 by using a SNP present in the transcribed region and found that this gene was transcribed on both parental alleles with similar ratio in the neoplastic and peri-neoplastic tissues (Figure 2b). Thus, IC2 methylation and KCNQ1OT1 imprinting were similarly relaxed in the neoplastic and peri-neoplastic tissues and KCNQ1 was increased in the tumor of our patient.”.
Figure 2 legend: “Expression analysis of KCNQ1OT1, KCNQ1 and CDKN1C in the peri-neoplastic and neoplastic tissues. (a) Total expression levels of the three genes obtained by RT-qPCR and (b) allele-specific expression levels of KCNQ1OT1 obtained by Sanger sequencing following Reverse transcriptase-PCR. In (b) the rectangle indicates the polymorphism to distinguish the two alleles.”.
Discussion line 467: “However, it is important to note that IC2 methylation and KCNQ1OT1 imprinting were similarly relaxed in the neoplastic and peri-neoplastic tissues and KCNQ1 and CDKN1C were not repressed in the tumor with respect to normal tissue in our patient (Fig. 2 and Fig S2). Methylation mosaicism among tissues and the presence of non-tumor cells in the tumor biopsy may in part explain this result.”.
We added also the paragraph of expression analysis in Material and Methods
“2.7 Expression analysis
RNA was extracted from neoplastic and peri-neoplastic tissues using TRIZOL reagent (Invitrogen, ThermoFisher Scientific, Italy), according to manufacturer’s protocol. The concentration was determined using a NanoDrop spectrophotometer (ThermoFisher Scientific, Italy). About 1.5 µg of RNA was treated with RNase-free DNase (Sig-ma-Aldrich, St. Louis, Missouri, Stati Uniti), and first-strand complementary DNA (cDNA) was synthesized using QuantiTect Reverse Transcription Kit (Qiagen-Italia, Milan, Italy) following the manufacturer’s protocol.
Quantitative PCR (qPCR) analysis was performed in a StepOnePlus RT-qPCR system (Applied Biosystem). Each reaction was carried out in duplicate in a 20µl final volume with 10µl of SYBR-Green supermix, 0.5µM of each primer and 5µl of cDNA. The initial denaturation for each run was for 10 minutes at 95°C. This was followed by 40 amplification cycles, of 95°C for 15 seconds, and for 1minute at 60° followed by 30 seconds at 72 °C. Relative expression levels were normalized using GAPDH as reference gene. Primers used are the following: KCNQ1OT1: For 5’-AGCCAGACAGAAGCCCAATA-3’, Rev 5’-TGGCCTAACATATCATCCCTCC-3’; KCNQ1: For 5'-AACACACAGAAGGGGACTGC-3', Rev5'-GCCTGTGATTCTCCACGTTT-3; CDKN1C: For 5'- AGAGATCAGCGCCTGAGAAG-3', Rev 5'-CACCTTGGGACCAGTGTACC-3’; GAPDH: For 5'-TCTCCTCTGACTTCAACAGCGACA-3', Rev 5'-CCCTGTTGCTGTAGCCAAATTCGT-3'. Allele-specific expression analysis of KCNQ1OT1 was performed by amplifying about 100 ng of cDNA. Sequencing of the SNP rs1076621, allowed us to distinguish the two alleles. PCR primers used same as for Real Time-qPC (RT-qPCR).”.
- Since the IC2 LoM is usually mosaic, mosaic ratios in the PBL and the peri-neoplastic tissue may be different. In fact, methylation levels were 14% in the PBL and 0.33 (33%) in the peri-neoplastic tissue. The authors hypothesized that the germline CFTR mutations was the first hit and the IC2 LoM was the second hit of the tumorigenesis. If his is the case, almost all tumor cells show the IC2 LoM, and methylation level of the neoplastic tissue may depend on percentage of tumor cells. However, the methylation levels at IC2 in the peri-neoplastic and neoplastic tissues were not so different. The authors should show the percentage of tumor cells in the neoplastic tissue, which they analyzed, and explain these methylation statuses.
R#2-2 We thank the reviewer for this interesting comment. According to the histological analysis report, the tumor cells represent 60% of cells in tumor biopsy (please, see also below, response R#2-4 to the first of the minor comments).
We have discussed this issue in Discussion section, at line 467, as follows:
“However, it is important to note that IC2 methylation and KCNQ1OT1 imprinting were similarly relaxed in the neoplastic and peri-neoplastic tissues and KCNQ1 and CDKN1C were not repressed in the tumor with respect to normal tissue of our patient (Fig. 2b and Fig S2). Methylation mosaicism among tissues and the presence of non-tumor cells in the tumor biopsy may in part explain this result.”.
Line 494: “We hypothesize that in our case the germline CFTR mutations represent the first hit of the tumorigenesis process and that this is accelerated by IC2 LoM and/or MLID.”.
- In the peri-neoplastic tissue, many iDMRs showed abnormal methylation. How the authors defined and found these abnormal methylations and why these abnormal methylations occurred in the peri-neoplastic tissue?
R#2-3 We considered affected the iDMRs showing methylation values that in the neoplastic tissue were at least 10% higher or lower than in the peri-neoplastic tissue. Because no further normal colorectal tissue specimen was available, we were unable to reveal any abnormal methylation that was present in the peri-neoplastic sample.
We have now added this information in the legend of Fig 3c as follows: “(c) Heat map showing the methylation profile of the iDMRs in neoplastic and peri-neoplastic tissues. The iDMRs with mean methylation difference of at least 10% between neoplastic and peri-neoplastic tissues are indicated in bold. Methylation changes possibly present in the peri-neoplastic tissue could not be detected as no normal colorectal tissue specimen was available for normalization.”.
Minor comments
- Detailed pathological information of the peri-neoplastic and neoplastic tissues should be described
R#2-4 The histological, immunohistochemical and mutational analyses were performed only on the neoplastic tissue by our pathology department. The results obtained from these analyses were already reported in the manuscript at the beginning of the Results paragraph. We have now added a few available information at line 220: “The reported histological diagnosis was: rectal mucosa showing neoplastic proliferation with features of invasive adenocarcinoma composed by 60% of tumor cells”.
- As the authors mentioned, CIMP was a well characterized molecular mechanism of CRC. It would be better to present the CIMP data.
R#2-5 According to Tapial et al. (2019, PMID 31324877), CRC is classified as CIMP-positive when the promoters of at least 5 of 8 biomarkers are hypermethylated in the tumor compared with the peri-neoplastic tissue. As no methylation difference was found at the promoter of any of the 8 biomarkers (CACNA1G, CDKN2A, CRABP1, IGF2, MLH1, NEUROG1, RUNX3, SOCS1), we considered this tumor CIMP-negative.
We have added this information in:
-Results line 341, as follows: “Also, 8 CIMP-specific genes revealed no methylation change at the CpG islands-promoters, thus this tumor was considered CIMP-negative (Ref Tapial et al., 2019) (Fig S3).”
-Legend to Fig S3: “Figure S3. Methylation analysis of the promoters of 8 CIMP-specific genes (CACNA1G, CDKN2A,CRABP1, IGF2, MLH1, NEUROG1, RUNX3 and SOCS1) according to Tapial et al., 2019. As the methylation levels were comparable between neoplastic and peri-neoplastic tissues, the tumor was considered as CIMP negative.”.
-Discussion line 436: “Also, normal methylation of the promoters of 8 CIMP-specific genes reveals the CIMP negative status of the tumor”.
- In the legend of Figure 4b, the domain names of CFTR protein should be spelled out.
R#2-6 The figure 4b (now Fig 5b) has been updated and the domain names spelled out in the legend. Please, see also answer R#1-5 to Reviewer#1.
- In Discussion, the author should discuss CDKN1C as well as KCNQ1OT1/KCNQ1.
R#2-7 We have discussed the role of CDKN1C in CRC by modifying the Discussion paragraph at line 480 as follows:
“Additionally, the lncRNA KCNQ1OT1 regulates the imprinted expression of the KCNQ1 and CDKN1C genes, both showing tumor suppressor activity in several tumors, including CRC. KCNQ1 encodes a potassium channel protein subunit, and represents an early prognostic biomarker in CRC, whose deficiency is associated with poor outcome [43]. It has been demonstrated that KCNQ1 limits oncogenesis by inhibiting nuclear localization of β-catenin and by maintaining adherent junctions that prevent epithelial to mesenchymal transition [44]. CDKN1C is an important inhibitor of the cell cycle, whose loss of expression appears associated with colorectal carcinogenesis and poor prognosis (Jia-Qing Li et al., 2003, PMID: 14612924 2003; Sun et al., 2011, PMID: 21278784).
- A SNP-array analysis revealed that non-mosaic duplications affected the imprinted loci of GNAS (chr 20q), PEG13 and ERLIN2 (chr 8q) in the neoplastic tissue. Although methylation levels of GNAS and PEG13 were altered, that of ERLIN2 was not altered (Figure 2C and Table S3). The authors should explain these findings.
R#2-8 Similar to IC2, the ERLIN2 DMR was hypomethylated in blood indicating a constitutional methylation defect (Fig.1b). Since aberrant methylation in the neoplastic tissue is defined from the comparison with the peri-neoplastic tissue, it is likely that we were not able to appreciate further methylation changes of this hypomethylated DMR due to the CNV. We have added this information in the text at line 345 as follows: “In particular, non-mosaic duplications and microduplications affected the long arm of chrs 8 and 20 and the chromosome regions including the oncogenes cMYC(chr8q24.21), KRAS (chr 12p12.1) and AURKA (20q13.2) and the imprinted loci GNAS (chr 20q), PEG13 and ERLIN2(chr 8q). Despite the CNV, no further methylation defect of ERLIN2 was detectable in the tumor (Figure 3c), likely because of the constitutional hypomethylation affecting this DNA region.”.
- It would be better to add pyrosequencing data of all iDMRs analyzed in the colorectal tissues to Figure S2.
R#2-9 We have updated Fig.S2 by including four additional iDMRs and Table S1 with the relative primers. The legend of Fig S2 has been modified as follows: “DNA methylation analysis by pyrosequencing. The methylation levels of 8 iDMRs were determined by sodium bisulfite DNA treatment, amplification and pyrosequencing in the neoplastic (dark cyan) and peri-neoplastic tissues (cyan). Consistent with the methylation-array results, the MEST:alt-TSS-DMR, GNAS-AS1:TSS-DMR, RB1:Int2-DMR and NDN:TSS-DMR showed differential methylation, while the H19/IGF2:IG-DMR, IGF2:alt-TSS-DMR, KCNQ1OT1:TSS-DMR and NNAT:TSS-DMR showed similar methylation levels between the neoplastic and peri-neoplastic tissues.”
Reviewer 3 Report
Dear authors, although this work is centered entirely on a single case putting the conclusions in doubt, I think it is useful to continue the research on other cases despite the difficulty of recruitment. Perhaps it could suggest studying epigenetic mechanisms in other rare diseases as well. The work was carried out in a well detailed and structured way despite its complexity.
Author Response
Point-by-point response to the reviewers’ comments and a description of the changes made in the manuscript CANCERS-2218673.
We thank the reviewers for their valuable comments on the original manuscript, CANCERS-2218673, “Co-occurrence of Beckwith–Wiedemann Syndrome and early-onset colorectal cancer”.
We have carefully reviewed the suggestions/concerns of the reviewers and addressed such points. Below we report point-by-point responses (in red) to the reviewers’ comments, and a description of the changes made in the manuscript (underlined).
REVIEWER# 3
Dear authors, although this work is centered entirely on a single case putting the conclusions in doubt, I think it is useful to continue the research on other cases despite the difficulty of recruitment. Perhaps it could suggest studying epigenetic mechanisms in other rare diseases as well. The work was carried out in a well detailed and structured way despite its complexity.
We thank the reviewer #3 for recognizing the relevance of our manuscript.
Reviewer 4 Report
Cecere, F et al. present in their manuscript “Co-occurrence of Beckwith–Wiedemann Syndrome and early- 2 onset colorectal cancer” a case report of a young female patient suffering from EO-CRC. The early occurrence is explained by two likely pathogenic germline CFTR mutation in combination with LOM at the KCNQ1OT1 DMR. The manuscript is well written, and the clinical features of the patients are comprehensively described. The conclusion that the both alterations contributing to the development of the EO-CRC are convincing and giving important implication for the clinical management of BWS patients.
Minor issues:
- Page 4, line 168: Should be “whole exome sequencing” and not “whole-genome exome sequencing”.
- Page7, line 275: Should be “PolyPhen-2”
Author Response
Point-by-point response to the reviewers’ comments and a description of the changes made in the manuscript CANCERS-2218673.
We thank the reviewers for their valuable comments on the original manuscript, CANCERS-2218673, “Co-occurrence of Beckwith–Wiedemann Syndrome and early-onset colorectal cancer”.
We have carefully reviewed the suggestions/concerns of the reviewers and addressed such points. Below we report point-by-point responses (in red) to the reviewers’ comments, and a description of the changes made in the manuscript (underlined).
REVIEWER# 4
Cecere, F et al. present in their manuscript “Co-occurrence of Beckwith–Wiedemann Syndrome and early- 2 onset colorectal cancer” a case report of a young female patient suffering from EO-CRC. The early occurrence is explained by two likely pathogenic germline CFTR mutation in combination with LOM at the KCNQ1OT1 DMR. The manuscript is well written, and the clinical features of the patients are comprehensively described. The conclusion that the both alterations contributing to the development of the EO-CRC are convincing and giving important implication for the clinical management of BWS patients.
We thank the reviewer #4 for appreciating the value of our manuscript.
Minor issues:
- Page 4, line 168: Should be “whole exome sequencing” and not “whole-genome exome sequencing”.
R#4-1 “whole-genome exome sequencing” has been replaced by “whole exome sequencing”
- Page7, line 275: Should be “PolyPhen-2”
R#4-2 Poliphen-2 has been replaced by PolyPhen-2.
Round 2
Reviewer 2 Report
The authors responded point-by-point to the reviewer’s comments. However, there is still two minor comments.
1. Primer sequences for expression analysis should be shown in supplemental table, like Table S1 and S2.
2. In Figure S3, the authors should explain what the different line colors indicate.
Author Response
The authors responded point-by-point to the reviewer’s comments. However, there is still two minor comments.
We thank the reviewer #2 for the useful comments.
- Primer sequences for expression analysis should be shown in supplemental table, like Table S1 and S2.
R#2-1 We have added Table S3 which lists the primers used for expression analysis.
- In Figure S3, the authors should explain what the different line colors indicate.
R#2-2 We have added the meaning of the line colors in the legend of Fig S3.